# Synergistic Antimicrobial Activity of Silver Nanoparticles with an Emergent Class of Azoimidazoles

**DOI:** 10.3390/pharmaceutics15030926

**Published:** 2023-03-13

**Authors:** Ana Isabel Ribeiro, Bárbara Vieira, Daniela Dantas, Bárbara Silva, Eugénia Pinto, Fátima Cerqueira, Renata Silva, Fernando Remião, Jorge Padrão, Alice Maria Dias, Andrea Zille

**Affiliations:** 1Centre for Textile Science and Technology (2C2T), Department of Textile Engineering, University of Minho, Campus of Azurém, 4800-058 Guimarães, Portugal; 2Department of Chemistry, Chemistry Centre of University of Minho (CQUM), University of Minho, Campus of Gualtar, 4710-057 Braga, Portugal; 3Laboratory of Microbiology, Biological Sciences Department, Faculty of Pharmacy, University of Porto, 4050-313 Porto, Portugal; 4UCIBIO—Applied Molecular Biosciences Unit, REQUIMTE, Laboratory of Toxicology, Department of Biological Sciences, Faculty of Pharmacy, University of Porto, Rua de Jorge Viterbo Ferreira n° 228, 4050-313 Porto, Portugal; 5Associate Laboratory i4HB—Institute for Health and Bioeconomy, Faculty of Pharmacy, University of Porto, 4050-313 Porto, Portugal; 6CIIMAR/CIMAR, Interdisciplinary Centre of Marine and Environmental Research, Terminal de Cruzeiros do Porto de Leixões, 4450-208 Matosinhos, Portugal; 7Molecular Oncology and Viral Pathology Group, Research Center of IPO Porto (CI-IPOP)/RISE@CI-IPOP (Health Research Network), Portuguese Oncology Institute of Porto (IPO Porto)/Porto Comprehensive Cancer Center (Porto.CCC), 4200-072 Porto, Portugal; 8Faculty of Health Sciences, Fernando Pessoa University, 4200-150 Porto, Portugal; 9Instituto de Investigação, Inovação e Desenvolvimento (FP-I3ID), Biomedical and Health Sciences Research Unit (FP-BHS), Universidade Fernando Pessoa, Praça 9 de Abril, 349, 4249-004 Porto, Portugal

**Keywords:** antifungal, antibacterial, synergism, silver nanoparticles, stabilizing agents

## Abstract

The combination of two or more agents capable of acting in synergy has been reported as a valuable tool to fight against pathogens. Silver nanoparticles (AgNPs) present a strong antimicrobial action, although their cytotoxicity for healthy cells at active concentrations is a major concern. Azoimidazole moieties exhibit interesting bioactivities, including antimicrobial activity. In this work, a class of recently described azoimidazoles with strong antifungal activity was conjugated with citrate or polyvinylpyrrolidone-stabilized AgNPs. Proton nuclear magnetic resonance was used to confirm the purity of the compounds before further tests and atomic absorption spectroscopy to verify the concentration of silver in the prepared dispersions. Other analytical techniques elucidate the morphology and stability of AgNPs and corresponding conjugates, namely ultraviolet–visible spectrophotometry, scanning transmission electron microscopy and dynamic light scattering analysis. The synergistic antimicrobial activity of the conjugates was assessed through a checkerboard assay against yeasts (*Candida albicans* and *Candida krusei*) and bacteria (*Staphylococcus aureus* and *Escherichia coli*). The conjugates showed improved antimicrobial activity against all microorganisms, in particular towards bacteria, with concentrations below their individual minimal inhibitory concentration (MIC). Furthermore, some combinations were found to be non-cytotoxic towards human HaCaT cells.

## 1. Introduction

The importance of antimicrobial agents in the medical sector is indisputable, either in treating or preventing infections, as well as in functionalizing biomedical materials and medical devices [1]. There is a high prevalence of multidrug-resistant (MDR) microorganisms, which increases the importance of the search for new effective agents, with special attention to the ones that avoid biofilm formation (which display a 10- to 1000-fold higher resistance). MDR pathogens are the main reason for chronic and healthcare-associated infections (HAIs), causing morbidity and mortality and having a significant economic and social burden [2,3]. Every year, MDR infections have been reported as the cause of around 700,000 deaths worldwide. This number has been estimated to increase to 10 million by 2050 [4,5]. Knowing that the development of new antimicrobial agents is expensive and time-consuming, the research for alternative antimicrobial strategies has been highly recommended to scientists [4]. Novel antimicrobial agents/strategies can have a special impact on: (i) acute, chronic, or postoperative wound care [6,7]; (ii) biomaterials for soft or hard implants [8,9]; (iii) the development of smart materials and sensors to detect or, in the case of infection, to immediately attack the pathogens [10]; and (iv) the coating materials needed to avoid surface contaminations [11]. These strategies may prevent HAIs, cross-contaminations, hospital admissions, amputations, patient discomfort and even death [12,13].

Antimicrobial agents with a broad-spectrum action against bacteria and fungi are useful in fighting polymicrobial infections/contaminations; they are also useful in empirical approaches, where it may not be possible to identify the pathogenic microorganism [14,15]. Many common infections are polymicrobial (e.g., surgical wounds or diabetic foot ulcers, otitis media, urinary tract infections, and lung infections in cystic fibrosis). The treatment of polymicrobial infections has additional therapeutic challenges and may have a considerable impact on the effectiveness of the antimicrobial agents [16]. Therefore, the broad-spectrum antimicrobials present a huge interest and have been used for several medical treatments (e.g., surgeries, premature infant care, organ transplantation, diabetics, and cancer chemotherapy) to treat or prevent opportunistic pathogens in immunocompromised patients [1]. Thus, broad-spectrum antimicrobial substances have been incorporated into pharmaceuticals, water filtration systems, fibers, packaging, surgical materials, and surfactants [17,18].

Nanomaterials are a promising area in biomedical sciences, including antimicrobial applications. In particular, silver nanoparticles (AgNPs) present antibacterial, antifungal, antiviral, and anti-inflammatory activity [19]. However, despite their vast benefits, there are still concerns associated with their application due to potential cytotoxic effects with long-term use of active concentrations. The cytotoxic effects of AgNPs in mammalian cells are strongly dependent on their physicochemical properties such as size, shape, surface charge, oxidation state, stabilization, agglomeration, and applied dosage [20,21,22].

In another field, for the purpose of producing more potent antimicrobial structures, a variety of heteroaromatic moieties have been studied. *N*-heterocycle scaffolds, such as imidazole rings, are readily available in nature and provide interesting biological functions such as antifungal, antibacterial, and anticancer activity [23,24,25,26,27,28]. In our research group, an unusual class of azo dyes containing 2-aminoimidazoles (AzoIz) was developed using 5-aminoimidazole 4-carboxamidrazones (Amz) as a precursor [29,30]. The initial Amz was oxidized using silver nitrate to obtain the AzoIz molecules. Then, the reaction of the AzoIz with secondary amines, namely dimethylamine (DMA) and piperidine (Pip), gave rise to a novel class of substituted AzoIz (AzoIz.DMA and AzoIz.Pip) with remarkable substituent patterns in the heteroaryl unit [30]. The antimicrobial activity of the neutral form of these molecules was screened against Gram-positive and Gram-negative bacteria, filamentous fungi, and yeasts. The compounds showed remarkable properties against *Candida* strains and against *Cryptococcus neoformans* and presented efficacy against biofilm formation. However, they were not effective against bacteria strains [29,30,31,32]. Moreover, the azoimidazole molecules also exhibited noteworthy halochromic properties, displaying strong colors that change from magenta to blue as the pH value decreases [30].

The conjugation of two or more antimicrobial agents acting in synergy may be a powerful tool to fight against pathogens, and several reports can be found in the literature combining silver nanoparticles and organic antimicrobial agents containing antibiotics, antifungals, and antivirals [33]. Despite all the intrinsic antimicrobial mechanisms of AgNPs (e.g., cell wall or cell membrane disruption increasing its permeability, cell wall depolarization, reactive oxygen species (ROS) generation, damaging of proteins and nucleic acids), they can also act as drug delivery systems. The conjugates can extend the circulation of the antimicrobials in the human body, accelerate their interaction with intercellular targets, and improve conjugate stabilization. At the same time, it is possible to reduce the required individual drug dosages, minimize toxicity, and broaden the antimicrobial spectrum [33,34,35]. Thus, the main goal of this work was to increase the spectrum of antimicrobial activity of these individual molecules by conjugating them with AgNPs to obtain a synergistic effect. Here, two types of stabilizing agents for AgNPs (citrate (Cit) and polyvinylpyrrolidone (PVP)) were tested in order to study their influence on the physicochemical properties of the conjugate and their corresponding antimicrobial activity.

## 2. Materials and Methods

AgNPs stabilized with Cit or PVP were used to compare the effect of the stabilizing agent with the physicochemical properties of AgNPs and, consequently, with the antimicrobial and cytotoxic results. Four imidazole-based molecules were selected from a synthetic pathway that involves the amidrazone precursor (Amz), the azoimidazole intermediate (AzoIz), and two of the final 2-aminoimidazole azo dyes (AzoIz.DMA and AzoIz.Pip). After the assays were performed on the Amz precursor, it was tested against the azoimidazole intermediate obtained from the Amz precursor through the direct oxidation of the amidrazone substituent. Then, the Amz precursor was evaluated against two products, AzoIz.DMA and AzoIz.Pip, which were prepared from the reaction of the AzoIz with DMA and Pip, respectively (Figure 1). The conjugates were analyzed by ultraviolet–visible (UV–vis) spectrophotometry to predict physicochemical modifications in AgNPs and/or organic molecules after the combination. Dynamic light scattering (DLS) analyses were performed to evaluate the stability of the AgNPs alone and conjugated by measuring the hydrodynamic size, polydispersity index (PdI), and zeta potential. Scanning transmission electron microscopy (STEM) was used to morphologically characterize the AgNPs. The antimicrobial activity was assessed against yeasts (*Candida albicans* and *Candida krusei*) and bacteria (*Staphylococcus aureus* and *Escherichia coli*) by the checkerboard method. The cytotoxicity of the nanoparticles alone and with conjugates was assessed using human HaCaT cells.

### 2.1. Materials

All the reagents used were of analytical grade or of the highest purity grade available. Diaminomaleonitrile (DAMN), 1,4-dioxane, triethyl orthoformate (TEOF), dimethylamine, and 1,8-diazabicyclo [5.4.0]undec-7-ene (DBU) were purchased from Acros Organics, Thermo Fisher Scientific, Alfagene, Portugal. Acetonitrile, silica gel, diatomaceus earth, methylamine, anilinium chloride, silver nitrate standard solution, and tri-sodium citrate dihydrate were acquired from Sigma Aldrich, Hamburg, Germany. Acetic acid was acquired from Chemlab, Zedelgem, Belgium; piperidine from Riedel-de Haen, North Carolina, US; phenylhydrazine and Macherey-Nagel™ aluminum sheets UV254 from Thermo Fisher Scientific, Alfagene, Portugal; diethyl ether, *n*-hexane, silica gel 60, silver nitrate, and sodium hydroxide were purchased from Panreac, Barcelona, Spain. Deuterated DMSO was purchased from TCI, Zwijndrecht, Belgium. Absolute ethanol and sodium borohydride were purchased from VWR chemicals, Carnaxide, Portugal. AgNPs.PVP 99.95%, with size 20–30 nm, were purchased from SkySpring Nanomaterials Inc., Houston, TX, USA.

Antimicrobial tests were performed using TSB for bacteria (Liofilchem, Roseto degli Abruzzi, Italy) and Sabouraud dextrose broth (SDB, Biokar, Pantin, France) for fungi growth; RPMI 1640 broth medium (Biochrom, Berlin, Germany) buffered to pH 7.0 with 3-(*N*-morpholino)propanesulfonic acid (MOPS; Sigma-Aldrich, St. Louis, MO, USA) was used for the checkerboard test.

Reagents used in the cell culture, including Dulbecco’s modified Eagle’s medium (DMEM) with 4.5 g·L^−1^ glucose and GlutaMAX™, heat-inactivated fetal bovine serum (FBS), 0.25% trypsin/1 mM ethylenediaminetetraacetic acid (EDTA), antibiotic (10,000 U·mL^−1^ penicillin, 10,000 µg·mL^−1^ streptomycin), and Hanks’ balanced salt solution (HBSS) without calcium and magnesium [HBSS (-/-)], were acquired from GibcoTM (Thermo Fisher Scientific, Alfagene, Portugal). Trizma^®^ base, neutral red (NR) solution, and resazurin (REZ) were obtained from Sigma-Aldrich (Merck KGaA, Darmstadt, Germany). Triton™ X-100 detergent solution and glacial acetic acid were acquired from Thermo Fisher Scientific (Waltham, MA, USA). Sulforhodamine B (SRB) was acquired from Santa Cruz Biotechnology (Dallas, TX, USA). Dimethyl sulfoxide (DMSO) was obtained from Merck (Darmstadt, Germany). Methanol (≥99.8%) was acquired from VWR Chemicals (Radnor, PA, USA). All sterile plastic materials used were acquired from Corning Costar (Glendale, AZ, USA).

### 2.2. Synthesis and Characterization of Amidrazone (Amz) and Azoimidazoles (AzoIz, AzoIz.DMA and AzoIz.Pip) Compounds

The antimicrobial molecules were prepared by well-known methods previously reported. The (*Z*)-5-amino-1-methyl-*N’*-phenyl-1*H*-imidazole-4-carbohydrazonamide (Amz) was prepared in a four-step synthetic pathway using the commercial reagents DAMN, TEOF, methylamine, and phenylhydrazine [29,36]. €-5-amino-4-(imino(phenyldiazenyl)methyl)-1-methyl-1*H*-imidazol-3-ium (AzoIz), (*E*)-4-(ami€(*E*)-phenyldiazenyl)methylene)-5-imino-*N,N,*1-trimethyl-4,5-dihydro-1*H*-imidazol-2-amine (AzoIz.DMA), and (*E*)-(5-imino-1-methyl-2-(piperidin-1-yl)-1,5-dihydro-4*H*-imidazol-4-ylidene)((*E*)-phenyldiazenyl)methanamine (AzoIz.Pip) were obtained from Amz and the commercial reagents silver nitrate and dimethylamine (DMA) or piperidine (Pip) through a two-step method presented in Figure 1 [30]. All compounds were characterized by ^1^H NMR (Avance 3400, Bruker, MA, USA) before further tests, and the corresponding peaks were compared with the pure products previously characterized [29,30].

### 2.3. Preparation of AgNPs Dispersions and Its Combination with Organic Molecules

All materials were cleaned with nitric acid (10% *v*/*v*) and rinsed with distilled water before the preparation of the AgNPs dispersions.

The AgNPs.Cit were synthesized according to the method described by Wan et al., with slight modifications [37,38]. Concisely, a seed dispersion (AgNPs with a size of 4 nm) was prepared by heating a citrate solution (1% *w*/*v*, 20 mL) with distilled water (75 mL) at 70 °C for 15 min. Subsequently, silver nitrate solution (1% *w*/*v*, 1.7 mL) was added, followed by a rapid addition of sodium borohydride solution (0.1% *w*/*v*, 2 mL) under vigorous stirring. The reaction was kept at 70 °C for 1 h. The final volume of the solution was adjusted to 100 mL using distilled water. The AgNP growth step was performed by boiling a citrate solution (1% *w*/*v*, 2 mL) and 35 mL of distilled water for 15 min. Then, starter seeds of 4 nm (10 mL) and silver nitrate (1% *w*/*v*, 1.7 mL) were added under vigorous stirring. The reaction was maintained at 70 °C for 1 h and, after cooling to room temperature, the volume was adjusted to 50 mL. Atomic absorption spectroscopy (AAS) was used to determine the Ag concentration in the final AgNPs.Cit dispersion, using a novAA^®^ 350 (Analytik Jena Analytical Instrumentation, Jena, Germany) after acid digestion. The final dispersion was diluted with distilled water to obtain the AgNPs.Cit with a final concentration of 200 μg·mL^−1^.

The AgNPs.PVP dispersion (200 μg·mL^−1^) was prepared by mixing the commercial nanoparticles in water using an ultrasonic bath for 30 min at 40 Hz (Model 3510, Branson, MO, USA) and an ultrasound tip for 15 min at 20 Hz (CY500, Optic Ivymen System, J.P Selecta, Barcelona, Spain). The final concentration of the AgNPs.PVP was also confirmed by AAS. For further analysis, the organic molecules (Amz, AzoIz, AzoIz.DMA, and AzoIz.Pip) were dissolved in water adding 2.0 molar equivalents of HNO_3_. Then, the molecules were conjugated with AgNPs.Cit and AgNPs.PVP in different concentrations according to the checkerboard test (described in Section 2.7).

### 2.4. UV–Vis Spectrophotometry

UV–vis spectra were collected in a Shimadzu UV-1800 (Shimadzu, Kyoto, Japan) using 1 cm wide quartz cells. The concentration of AgNPs and organic molecules (Amz, AzoIz, AzoIz.DMA, and AzoIz.Pip) in each solution was 12.5 μg·mL^−1^. The spectra of the solutions were recorded immediately after the preparation of solutions/dispersions and after 5, 30, and 60 min. As an exception, due to the high reactivity of the Amz molecule in the presence of AgNPs.Cit, the UV–vis spectra for this solution were collected after 1, 5, 15, 20, 30, 60, and 90 min.

### 2.5. Scanning Transmission Electron Microscopy (STEM)

Morphological analyses were carried out with an ultra-high-resolution Field Emission Gun SEM (FEG-SEM) NOVA 200 Nano (FEI Company, Hillsboro, OR, USA). Secondary electron images were performed with an acceleration voltage of 5 kV. Secondary electron images were obtained with an acceleration voltage from 5 to 17.5 kV. Backscattering electron images were realized with an acceleration voltage of 15 kV. Samples were coated with an Au-Pd (80–20 weight %) film using a high-resolution sputter coater (208HR, Cressington Company, Watford, UK) coupled to an MTM-20 Cressington High-Resolution Thickness Controller.

### 2.6. Dynamic Light Scattering (DLS) Analysis

The hydrodynamic size distribution, polydispersity index (PdI), and zeta potential of the AgNPs.Cit and AgNPs.PVP alone in the dispersion and conjugated with the organic molecules were measured using a Zeta Sizer-Nano (Malvern Instruments, Malvern, UK). Data were collected after 30 scans at 25 ± 1 °C, and zeta potential was measured in a moderate electrolytic concentrated solution. The presented values were obtained by averaging measurements of three samples.

### 2.7. Synergy Testing by Checkerboard Assay

The synergistic effect between the organic molecules and the AgNPs was tested against microorganisms purchased at American Type Culture Collection (ATCC; Manassas, VA, USA). The following microorganisms were tested: bacteria *Staphylococcus aureus* ATCC 25923 and *Escherichia coli* ATCC 25922; yeasts *Candida albicans* ATCC 10231 and *Candida krusei* ATCC 6258. The pre-inoculum of each bacterium was prepared in TSB and yeasts in SDB. After 12 h of incubation at 37 °C for bacteria and 26 °C for yeasts, and 120 rpm, the inoculum of each microorganism was centrifuged, the supernatant was eliminated, and the microorganism was resuspended with sterile NaCl (0.9%) solution. Then, the concentration of each microorganism was adjusted to 5 × 10^5^ CFU·mL^−1^.

Subsequently, the in vitro synergistic activity of the AgNPs.Cit and AgNPs.PVP combined with the Amz, AzoIz, AzoIz.DMA, and AzoIz.Pip was determined by a checkerboard assay [39]. Briefly, the checkerboard assay was conducted in 96-well microtiter plates divided into two groups considering columns. The rows contained the organic molecules with serial dilution concentrations varying from 128.0 μg·mL^−1^ to 2.0 μg·mL^−1^ (from rows A to G). The columns contained the AgNPs.PVP or AgNPs.Cit with serial dilution concentrations varying from 25.0 μg·mL^−1^ to 1.6 μg·mL^−1^ (from columns 6 to 2 or from columns 12 to 8). Thus, from columns 1 to 6 and from columns 7 to 12, two independent tests were performed. From the data, it was also possible to calculate the Minimal Inhibitory Concentration (MIC) for each compound or AgNP type alone, when the MIC value was found in the range of tested concentrations. The MIC of organic molecules could be found in columns 1 or 7, and the MIC of AgNPs in row H (Figure 2). In cases where the MIC was not among the tested concentrations, the value was searched in the literature. The plates were inoculated with the previously prepared inoculum and incubated at 37 °C for bacteria and 26 °C for yeasts for 24 h. At the same time, two 96-well plates were prepared under exactly the same conditions, but without microorganism inoculum, and were incubated at 37 °C and 26 °C.

The MICs of each antimicrobial agent alone and in combination were observed by naked eyes in order to measure the optical density (OD) at 600 nm for bacteria and 640 nm for yeasts. The data are reported in terms of the microorganism growth percentage calculated as Equation (1) and the fractional inhibitory concentration index (FICI) calculated following Equation (2). The MIC was defined as the smallest concentration of drug needed to reduce microorganism growth by more than 80% [39]. The FICI value was interpreted as synergistic (≤0.5), additive (>0.5 and ≤1.0), indifferent (>1.0 and ≤4.0), or antagonistic (>4.0) [40].
(1)$$Microorganism\;growth\;(\%)=\frac{OD\;agent\;combination\;well-OD\;background}{OD\;agent\;free\;well-OD\;background}$$
(2)$$FICI=\frac{MIC\;of\;agent\;A\;in\;combination}{MIC\;of\;the\;agent\;A\;alone}+\frac{MIC\;of\;agent\;B\;in\;combination}{MIC\;of\;the\;agent\;B\;alone}$$

### 2.8. Cytotoxicity

#### 2.8.1. HaCaT Cell Culture

The immortalized human keratinocyte (HaCaT) cell line was obtained from Cell Lines Service (CLS) (Eppelheim, Germany), and cells were routinely cultured in 75 cm^2^ flasks using DMEM with 4.5 g·L^−1^ glucose and GlutaMAX™, supplemented with 10% heat-inactivated FBS, 100 U·mL^−1^ penicillin, and 100 μg·mL^−1^ streptomycin. Cells were maintained in a 5% CO_2_–95% air atmosphere at 37 °C, and the medium was changed every 2–3 days. Cultures were passed weekly by trypsinization (0.25% trypsin/1 mM EDTA).

#### 2.8.2. Compound Cytotoxicity

The compound cytotoxicity was evaluated, 24 h after exposure, by the neutral red (NR) uptake, resazurin (REZ) reduction, and sulforhodamine B (SRB) binding assays. For that purpose, the cells were seeded in 96-well plates at a density of 60,000 cells.cm^−2^ and, 24 h after seeding, the cell culture medium was removed, and the cells were exposed to the tested compounds. A work solution of each compound was freshly prepared in cell culture medium and diluted, in cell culture medium, to obtain the desired concentrations. Triton™ X-100 (1% (*v*/*v*)) was used as positive control.

#### 2.8.3. Neutral Red Uptake Assay

Cytotoxicity of the compounds was evaluated by the NR uptake assay, in which the estimation of the number of viable cells in culture was assessed based on their ability to incorporate and bind the supravital dye NR into the lysosomes. At the selected time point (24 h), the cell culture medium was removed and replaced by fresh cell culture medium containing 50 μg·mL^−1^ NR. The cells were then incubated at 37 °C in a humidified 5% CO_2_–95% air atmosphere for 40 min. After the incubation period, the cell culture medium was removed, followed by the extraction of the dye absorbed only by viable cells with absolute ethyl alcohol/distilled water (1:1) with 5% (*v*/*v*) acetic acid. The absorbance was then measured at 540 nm in a multiwell plate reader (PowerWaveX BioTek Instruments, Winooski, VT, USA). The percentage of NR uptake relative to control cells (0 µg·mL^−1^) was used as the cytotoxicity measure. Results were obtained from four independent experiments, performed in triplicate.

#### 2.8.4. Resazurin Reduction Assay

Cytotoxicity of the compounds was further evaluated by the REZ reduction assay, in which the cellular metabolic capacity and cytotoxicity was assessed based on the ability of living cells to reduce the oxidized blue dye (REZ) to a fluorescent pink resorufin product. At the selected time point (24 h), the cell culture medium was removed and replaced by fresh cell culture medium containing 10 μg·mL^−1^ REZ. The cells were then incubated at 37 °C in a humidified 5% CO_2_–95% air atmosphere for 40 min. After the incubation period, the resorufin fluorescence was measured at excitation/emission wavelengths of 560/590 nm in a multiwell plate reader (PowerWaveX BioTek Instruments, Winooski, VT, USA). The percentage of REZ reduction relative to control cells (0 µg·mL^−1^) was used as the cytotoxicity measure. Results were obtained from four independent experiments, performed in triplicate.

#### 2.8.5. Sulforhodamine B Assay

Cytotoxicity of the compounds was also evaluated by the SRB binding assay, in which the number of cells (according to the total protein mass) was assessed based on the binding of the SRB dye to the basic amino acids of cellular proteins under mild acidic conditions. At the selected time point (24 h), the cell culture medium was removed, the cells were washed with HBSS (+/+) and fixed overnight at −20 °C with a methanolic solution of 1% acetic acid (*v*/*v*). Afterwards, the fixation medium was removed, and the cells were incubated with a 0.05% (*w*/*v*) SRB solution [prepared in 1% (*v*/*v*) acetic acid] for 45 min at 37 °C. At the end of this incubation period, the SRB solution was aspirated, and the cells were washed three times with 1% (*v*/*v*) acetic acid to ensure the complete removal of the unbound dye. The SRB bound to the basic amino acids of cellular proteins was then extracted with a Tris base solution (10 mM, pH 10.5). The absorbance was measured at 540 nm in a multiwell plate reader (PowerWaveX BioTek Instruments, Winooski, VT, USA). The percentage of SRB binding relative to the control cells (0 µg·mL^−1^) was used as the cytotoxicity measure. Results were obtained from four independent experiments, performed in triplicate.

#### 2.8.6. Statistical Analysis

All the statistical assessments were performed using the GraphPad Prism 8 for Windows (GraphPad Software, San Diego, CA, USA). The normality of data distribution was calculated using the KS, D’Agostino & Pearson omnibus, and Shapiro–Wilk normality tests. One-way ANOVA was used to perform the statistical comparisons, followed by the Dunnett’s multiple comparisons test. In all experiments, *p* values under 0.05 were considered statistically significant.

## 3. Results and Discussion

### 3.1. Conjugation of Organic Molecules with AgNPs and UV–Vis Characterization

The AgNPs.Cit were synthesized following a previously reported method that combines a seed-mediated growth and the Lee–Meisel process (thermal reduction of AgNO_3_ with citrate) [37]. The synthesis was confirmed by the color shift of the silver nitrate solution from colorless to dark yellow after adding the reducing agents in both the seed formation and the growth phase, and this synthesis was assigned to the reduction of silver ions to metallic silver. The UV–vis spectra of the AgNPs.Cit with the size of 20 nm displayed the presence of one absorption band at 392 nm, corresponding to the surface plasmon resonance (SPR) of the AgNPs.Cit that is consistent with the reported synthesis (Figure 3). The AgNPs.PVP were commercially obtained and redispersed in water. In tested concentrations, it was not possible to observe any UV–vis band. The organic molecules also displayed typical absorption bands in ultraviolet and visible regions (Figure 3). The Amz showed only one band in the UV region at 296 nm and, consequently, the initial solution was colorless. On the contrary, AzoIz solution exhibited an intense red color as, in addition to the presence of two bands in the UV region (277 and 334 nm), the molecule displayed one band in the visible region at 497 nm. The last two bands seemed to correspond to the typical bands observed in azobenzene dyes [41]. Similarly, AzoIz.DMA and AzoIz.Pip also showed three UV–vis bands, one at 287 nm, another one at 334–342 nm, and the last one at 560 nm. Thus, these molecules exhibited a strong violet color (Figure 3).

Using UV–vis data, it was possible to detect some molecular and AgNP modifications after the conjugation. Combining Amz with AgNPs.Cit, the oxidation of the molecule and the AgNP agglomeration was evidenced by the decrease in the peak attributed to AgNPs.Cit and the emergence of another peak at a higher wavelength. As time passed, absorption bands at 274, 347 and 500 nm arose, which correspond to the spectrum of AzoIz. This result suggested that complete oxidation of the Amz to AzoIz occurred, as it happened in the synthesis of the compound AzoIz. In addition, a stable red color also appeared after 60 min, confirming the presence of the AzoIz. When the AzoIz compound was combined with AgNPs.Cit, only a slight bathochromic shift was observed from 497 to 519 nm. In addition, the intensity of the AgNP peak decreased, indicating some possible agglomeration. Then, the conjugation of AgNPs.Cit with AzoIz.DMA and AzoIz.Pip did not show significant differences in the spectra with the passage of time. Only the band attributed to silver decreased, suggesting some AgNP agglomeration. In contrast, the conjugation of AgNPs.PVP did not change the absorption spectra of the compounds, maintaining similar UV–vis spectra, and color changes were observed (Figure 3). These perceived differences can be related to the stability of AgNPs.Cit vs. AgNPs.PVP. Charged stabilized nanoparticles (such as Cit) tend to be more unstable than sterically stabilized ones (such as PVP). Cit stabilizes the AgNPs by charge repulsion and is weakly bound to the silver core. PVP stabilizes the NPs sterically and is strongly bound to the core, being permeable to solutes and solvents [42,43].

### 3.2. Morphological Analysis of AgNPs Alone and Conjugated with Organic Molecules by STEM

STEM analysis was performed to evaluate the morphological characteristics of AgNPs under the different conditions tested. AgNPs.Cit showed well-dispersed quasi-spherical nanoparticles with an average diameter size of 21.2 ± 3.2 nm. However, the size of AgNPs.PVP was more difficult to measure since the PVP polymer surrounded the nanoparticles. Independently of this, the STEM images of AgNPs.PVP suggested some agglomeration with an average size of 73.4 ± 13.7 nm (Figure 4). After the conjugation procedure, no significant differences were observed in the size of the AgNPs.Cit using the different organic molecules, where the addition of AzoIz.DMA presented an average size of 21.4 ± 3.3 nm, and the addition of AzoIz.Pip showed an average of 17.9 ± 2.7 nm. In terms of agglomerations, AgNPs.Cit seemed to be better dispersed in the presence of AzoIz.DMA or AzoIz.Pip (Figure 4). Analyzing AgNPs.PVP after adding the organic molecules, the size seemed to slightly decrease to 44.7 ± 8.8 nm by the addition of AzoIz.DMA and to 37.8 ± 7.3 using AzoIz.Pip; however, some agglomerates are always noticeable.

### 3.3. DLS Measurements of AgNPs Alone and Conjugated with Organic Molecules

The aqueous solutions of AgNPs and their conjugates with organic molecules were characterized by DLS in different conditions to estimate the tendency of their stability according to molecule concentration. Thus, the maximum concentration of AgNPs was chosen (25.0 μg·mL^−1^) for all analyses, and two middle values of the tested concentrations of organic molecules were combined (16.0 or 32.0 μg·mL^−1^). The results of the AgNPs alone showed a hydrodynamic size of 34.89 ± 1.3 nm for AgNPs.Cit and a superior value for AgNPs.PVP (273.2 ± 1.6 nm), confirming some agglomeration in PVP-stabilized AgNPs as suggested by STEM images. The zeta potential value also showed greater dispersion stability in AgNPs.Cit (−30.5 ± 0.1 mV) than in AgNPs.PVP (−22.6 ± 0.1 mV) that could be related to the preparation method of the dispersions, once the AgNPs.Cit were directly synthesized and used, while the AgNPs.PVP dispersion was dried and nanoparticles were further redispersed. The conjugates of AgNPs.Cit showed a general decrease in the hydrodynamic size by increasing the concentration of the organic molecules. Even so, the larger concentration as tested (32.0 μg·mL^−1^) was not sufficient for maintaining the stability of the nanoparticles at the initial values of size and zeta potential, inducing some agglomeration. When AgNPs.PVP were used in the conjugates, the size and zeta potential values also exhibited some agglomeration. In this case, by increasing the concentration of organic molecules, the stability of the dispersion showed a general decrease (Table 1).

### 3.4. MIC Determination and Antibacterial Combination Assay (Checkerboard Results)

The antimicrobial efficiency of AgNPs (citrate or PVP stabilized) containing combinatorial organic molecules (Amz, AzoIz, AzoIz.DMA and AzoIz.Pip) in different concentrations (AgNPs from 25.0 to 1.6 μg·mL^−1^, and organic molecules from 128.0 to 2.0 μg·mL^−1^) were tested against bacteria (*S. aureus* and *E. coli*) and yeasts (*C. albicans* and *C. krusei*) using the checkerboard method. The MIC of each antimicrobial agent was directly tested on the checkerboard assay, needed as control results. The MIC obtained for organic molecules showed high values against bacteria (from 64.0 to 128.0 μg·mL^−1^) and low values against fungi (from 4.0 to 16.0 μg·mL^−1^) (Table 2). It is important to note that these tests were performed with the protonated form of the molecules, which slightly increases some MIC values, compared with the values reported using the neutral form [30]. The MIC for AgNPs was found to be above the concentration tested in the checkerboard assay. Thus, the MIC reported in the literature was used for FICI calculations (Table 2).

In previous work, the cytotoxicity of the organic molecules was tested alone revealing no cytotoxicity in concentrations smaller than or equal to 16.0 μg·mL^−1^ in immortalized human keratinocytes (HaCaT cell line) [30]. In this work, the checkerboard assay was used to evaluate the synergistic activity from 2.0 to 128.0 μg·mL^−1^ in order to find synergistic conditions against bacteria. Some results showed synergism above the values found in cytotoxicity tests of the molecules (16.0 μg·mL^−1^). However, only synergistic results with organic molecule concentrations smaller than or equal to 16.0 μg·mL^−1^ are presented in the manuscript. The other tested concentrations can be found in the supporting information with the indication of antimicrobial synergistic and additive results (Appendix A).

In general, the synergistic effect was more noticeable against bacteria, which can be justified by the inferior activity of the compounds against bacteria when compared with the yeasts. The checkerboard results were divided by organic molecules (Figure 5, Figure 6, Figure 7 and Figure 8) and considered synergism when two conditions were found in conjugates: (i) a microorganism growth percentage of ≤20% and (ii) FICI of ≤0.5.

By conjugating Amz with AgNPs, the FICI calculation was always less than or equal to 0.5, but only some combinations decreased the bacterial growth to less than 20%. It was possible to have a strong synergism in concentrations significantly smaller than the MIC for *S. aureus*, decreasing the concentration of Amz from 128 to 16.0 μg·mL^−1^ with a very small concentration of AgNPs–1.6 μg·mL^−1^. The concentration of Amz could still be reduced to 2.0 μg·mL^−1^, but larger concentrations of AgNPs were needed (25.0 μg·mL^−1^). Similarly, in intermediate combinations (8.0 and 4.0 μg·mL^−1^) some synergistic results emerged by combining larger concentrations of AgNPs.Cit (12.0 or 25 μg·mL^−1^) (Figure 6), which could be of interest depending on the cytotoxicity results as presented in Figure 9. When the stabilizing agent of AgNPs was the PVP polymer, no synergism was detected using non-cytotoxic concentrations. There were synergistic and additive results using larger Amz concentrations (32.0 or 64.0 μg·mL^−1^) (Appendix A).

Against *E. coli*, an evident synergism existed using AgNPs.Cit and AgNPs.PVP. The activity was effective using small concentrations of Amz (16.0, 8.0, 4.0 and 2.0 μg·mL^−1^) with small–medium amounts of AgNPs (1.6, 3.1, 6.1 and 12.5 μg·mL^−1^) (Figure 5).

Despite the low MIC value already obtained against *C. albicans*, a synergistic effect could be detected using even small concentrations (2.0 μg·mL^−1^ of Amz with 1.6 μg·mL^−1^ of AgNPs.Cit). Against *C. krusei*, synergism was detected throughout. Arguably, among all the organic molecules, Amz was the one that presented the best results in terms of antimicrobial synergy with AgNPs and showed a greater effect when citrate was used to stabilize the AgNPs. This could be justified by the superior agglomeration of AgNPs.PVP and by the amidrazone reactivity observed in UV–vis spectra when the AgNPs.Cit were conjugated.

Only AzoIz acted in synergy with AgNPs.Cit against *E. coli* as it was observed that 16.0 μg·mL^−1^ of AzoIz and 6.3 μg·mL^−1^ of the AgNPs.Cit inhibited *E. coli* growth. It was also observed that smaller concentrations of AzoIz could have a similar inhibition, but larger concentrations of AgNPs.Cit would be needed (Figure 6). Against yeasts, synergistic results were found, but some additive effects could be detected against *C. albicans* and *C. krusei*. The fungal growth percentage was very low (less than 20%) when AzoIz at 2.0 μg·mL^−1^ was combined with small concentrations of AgNPs.Cit (1.6–3.1 μg·mL^−1^) (Appendix A).

AzoIz.DMA and AzoIz.Pip showed few and similar synergistic results. Against *S. aureus*, the conjugates with 16.0 μg·mL^−1^ of the organic molecule with AgNPs.Cit 25.0 μg·mL^−1^ presented a suitable antibacterial action (Figure 7 and Figure 8). Against yeasts, only AzoIz.DMA showed some synergy against *C. albicans*, using 2.0 μg·mL^−1^ of AzoIz.DMA with 1.6–3.1 of AgNPs. Several additive results could be found against tested *Candida* strains (Appendix A).

Some works have described the interactions between AgNPs and antimicrobial agents. The formation of chelation bonds has been considered one of the most important interactions as it increases the concentration of antimicrobial agents on the cell membrane, enabling AgNPs to act as drug carriers. AgNP conjugates can also react with DNA, resulting in unwound DNA duplexes. Then, the enzymes responsible for drug hydrolysis can be inhibited by AgNPs, maximizing the activity of the antimicrobial agents. Moreover, the charge of AgNPs also presents a significant role. Positive AgNPs present better antimicrobial action, which facilitates their binding to the negatively charged surface of the bacteria [33,46]. Here, the conjugation of organic molecules with AgNPs was shown to decrease the initial negative zeta potential of AgNPs. This justifies the positive results obtained by using AgNPs conjugated with the organic molecules. Better results could likely be found if the agglomeration problems were avoided.

In general, the synergistic behavior of the organic molecules and AgNP conjugates could be attributed to the different effects on cellular targets that can complement each other. Amz molecules showed better results (Table 3) that can be related to the reactive oxygen species (ROS) formed in lower concentrations, as already studied by the research group [32]. However, further studies should be performed to better understand the mechanism of action of the conjugates.

### 3.5. Cytotoxicity Evaluation

Since the cytotoxicity of organic molecules towards HaCaT cells has already been studied in previous works (and which were defined as non-cytotoxic for concentrations equal to or smaller than 16.0 µg·mL^−1^) [30], in this work the AgNPs and the conjugates of organic molecules and AgNPs were tested in concentrations that were found to be synergistic (Figure 9). Concerning the AgNPs alone, it is possible to report no significant cytotoxic effects in the maximum concentrations used in this work (25.0 µg·mL^−1^). Significant cytotoxic effects were only detected when tested at 50.0 µg·mL^−1^, for both AgNPs.Cit and AgNPs.PVP. In general, the combination of the organic molecules with AgNPs did not significantly affect the viability of the cells (values > 88% vs. control; *p* > 0.05%), with only a few exceptions. A significant decrease in cell viability was observed in all three cytotoxicity assays for AzoIz.Pip when combined with AgNPs.Cit in the concentrations of 16.0 and 25.0 µg·mL^−1^, respectively. Using citrate-stabilized nanoparticles, the sulforhodamine B binding assay also showed a significant reduction in the cell viability of the Amz/AgNPs.Cit 8.0 + 1.6 µg·mL^−1^ conjugate. Using the AgNPs.PVP, the cytotoxic effects were more perceived using Amz in a concentration equal or superior to 8.0 µg·mL^−1^ as assessed by the resazurin reduction and sulforhodamine B binding assays. The neutral red uptake assay was less sensitive in addressing the cytotoxicity of the developed conjugate, with only the concentration of 16.0 µg·mL^−1^ significantly compromising cell viability. Furthermore, for all the dispersions of AgNPs.Cit and AgNPs.PVP with AzoIz and AzoIz.DMA, no remarkable cytotoxicity was detected. However, the two nanoparticles (AgNPs.Cit and AgNPs.PVP) had significant cytotoxicity when formulated at 25.0 µg·mL^−1^ in combination with the largest concentration of AzoIz.Pip (16.0 µg·mL^−1^), as evaluated by the three performed cytotoxic assays.

## 4. Conclusions

In this study, amidrazone and azoimidazole compounds were conjugated with Cit- or PVP-stabilized AgNPs. The physicochemical properties of individual agents and conjugates were assessed using microscopic and spectroscopic methods. The conjugates exhibited enhanced efficacy, especially against bacteria, acting in synergy once the applied amidrazone and azoimidazoles were intrinsically active against *Candida* strains. In general, synergism reduces the MIC for bacteria. MIC values for yeasts can even be reduced in some conjugates, particularly when the amidrazone molecule is used. In addition, AgNPs.Cit exhibited better antimicrobial action when combined with organic molecules, which was attributed to the higher stability and lower agglomeration of the dispersions when compared with AgNPs.PVP. The obtained results suggest that conjugates may represent a useful strategy to broaden the spectrum of the antimicrobial molecules studied. Thus, this unconventional strategy shows that it might be helpful to use these imidazole derivatives to fight against HAIs and polymicrobial infections or to functionalize biomedical materials or devices.

## Figures and Tables

**Figure 1 pharmaceutics-15-00926-f001:**
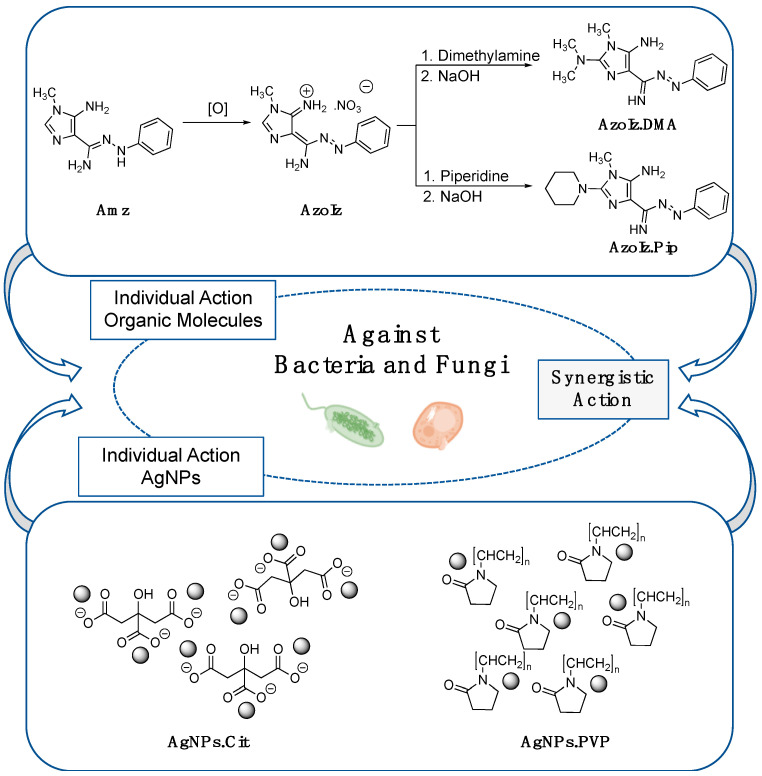
Schematic representation of the methodology adopted to obtain Amz/AzoIz and AgNPs conjugates.

**Figure 2 pharmaceutics-15-00926-f002:**
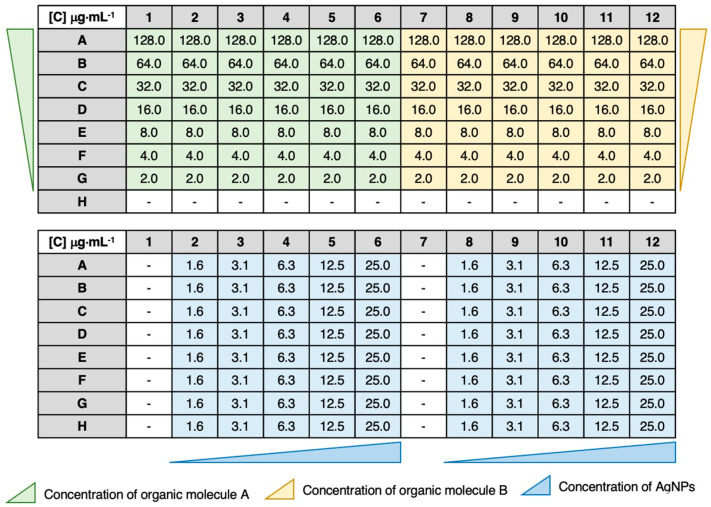
Schematic representation of the final concentration distribution in each 96-well microtiter plate in checkerboard assay. First, the organic molecules (Amz, AzoIz, AzoIz.DMA, or AzoIz.Pip) were serially diluted (green and yellow), and then, in the same plate, the AgNPs (stabilized in citrate or PVP) were added and serially diluted (blue). MIC of organic molecules can be calculated from columns 1 and 7, and MIC of AgNPs can be calculated twice from raw H.

**Figure 3 pharmaceutics-15-00926-f003:**
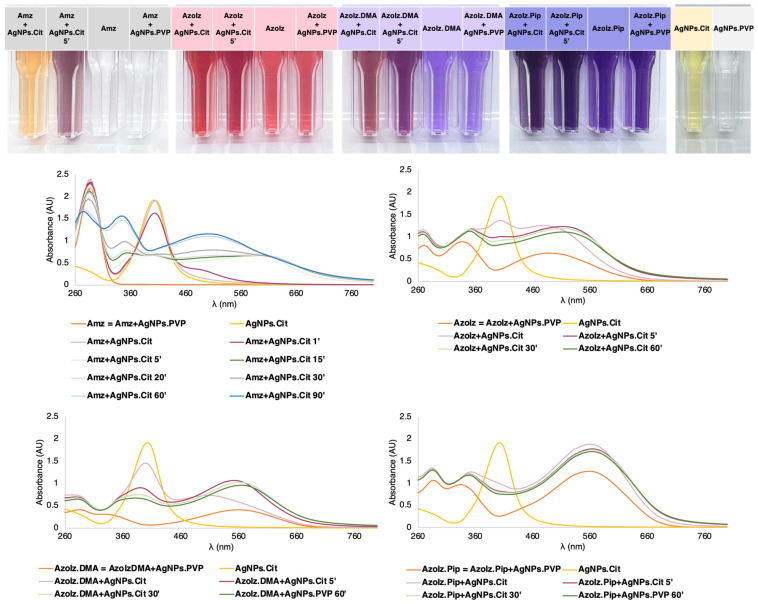
Colors of the dispersions after conjugation of Amz/AzoIz with AgNPs, corresponding controls, and UV–vis spectra.

**Figure 4 pharmaceutics-15-00926-f004:**
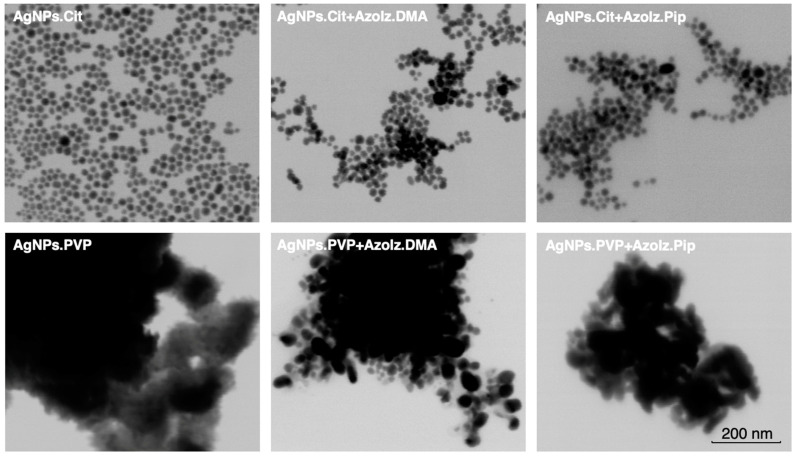
STEM images of AgNPs alone and combined with AzoIz.DMA or AzoIz.Pip at a magnification of 400,000×.

**Figure 5 pharmaceutics-15-00926-f005:**
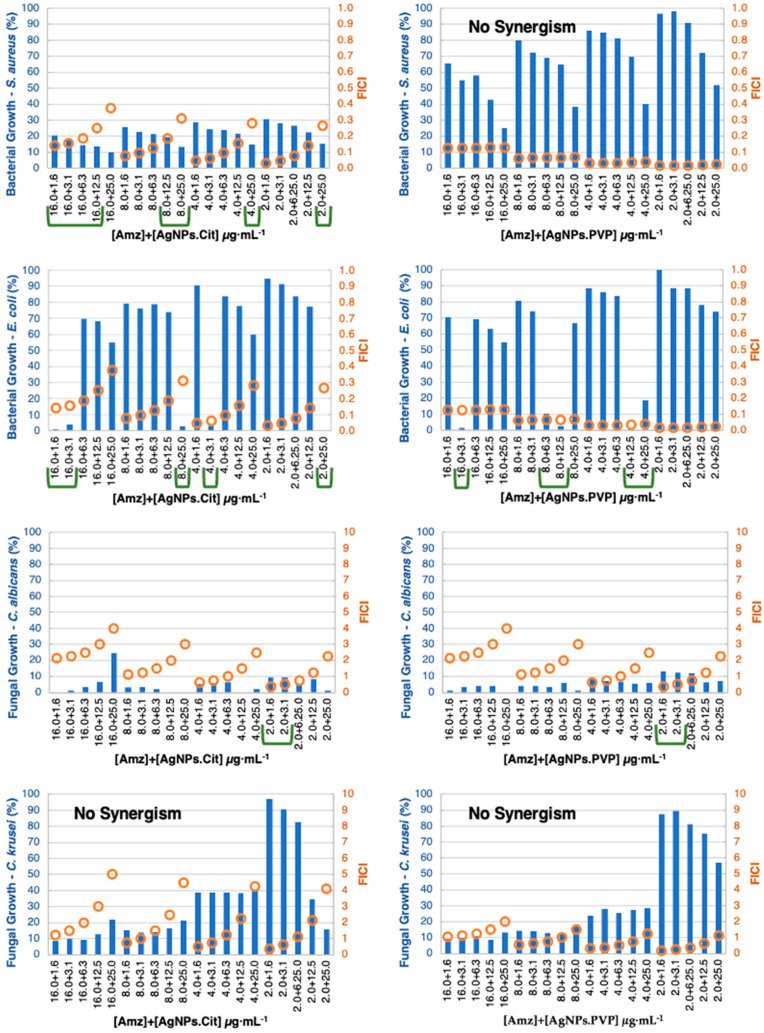
Checkerboard results by combining Amz (concentration ≤ 16.0 μg·mL^−1^) with AgNPs (concentration ≤ 25.0 μg·mL^−1^) showing positive synergistic results (green lines). Bars (blue) represent the microbial growth percentage, and dots (orange) represent the FICI values.

**Figure 6 pharmaceutics-15-00926-f006:**
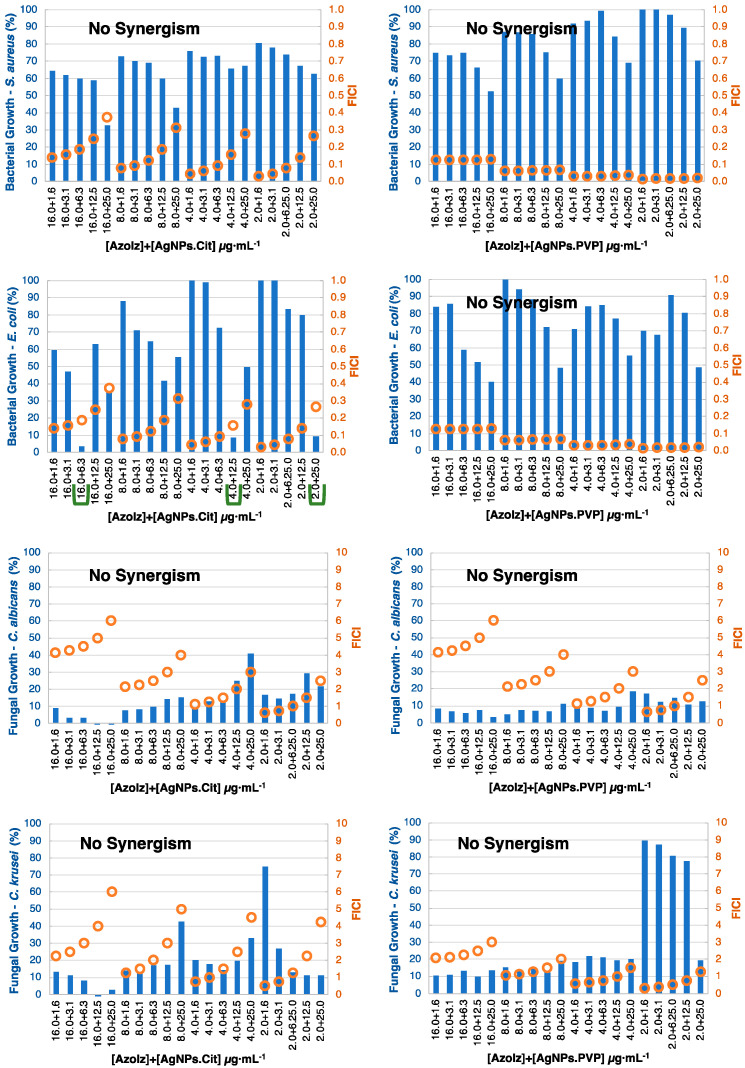
Checkerboard results by combining AzoIz (concentration ≤ 16.0 μg·mL^−1^) with AgNPs (concentration ≤ 25.0 μg·mL^−1^) showing positive synergistic results (green lines). Bars (blue) represent the microbial growth percentage, and dots (orange) represent the FICI values.

**Figure 7 pharmaceutics-15-00926-f007:**
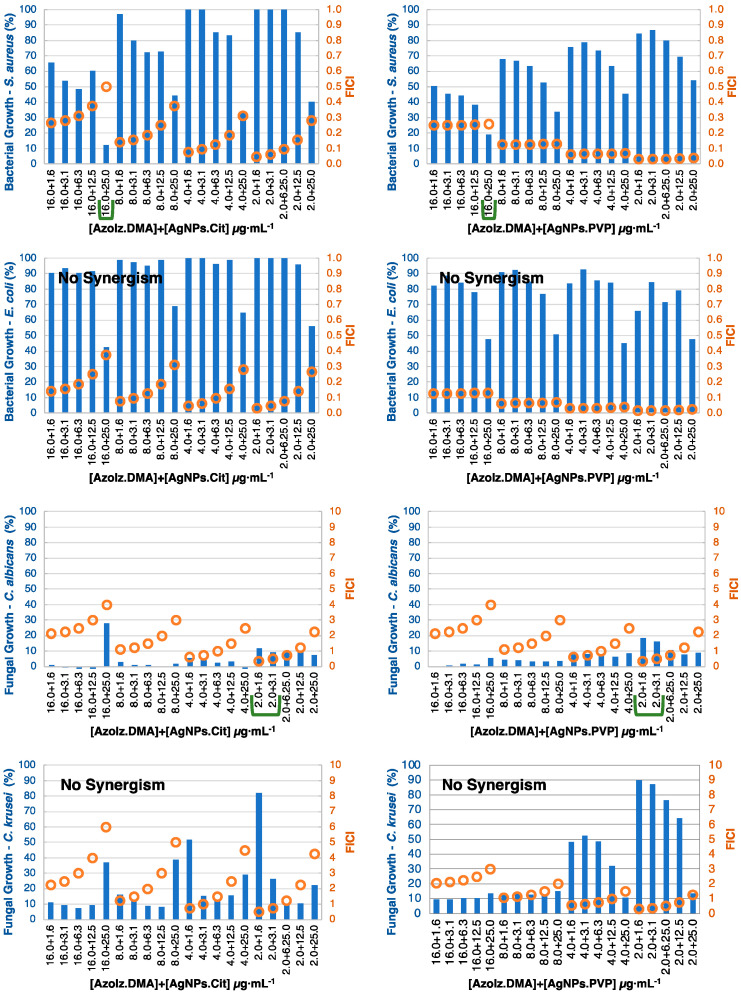
Checkerboard results by combining AzoIz.DMA (concentration ≤ 16.0 μg·mL^−1^) with AgNPs (concentration ≤ 25.0 μg·mL^−1^) showing positive synergistic results (green lines). Bars (blue) represent the microbial growth percentage, and dots (orange) represent the FICI values.

**Figure 8 pharmaceutics-15-00926-f008:**
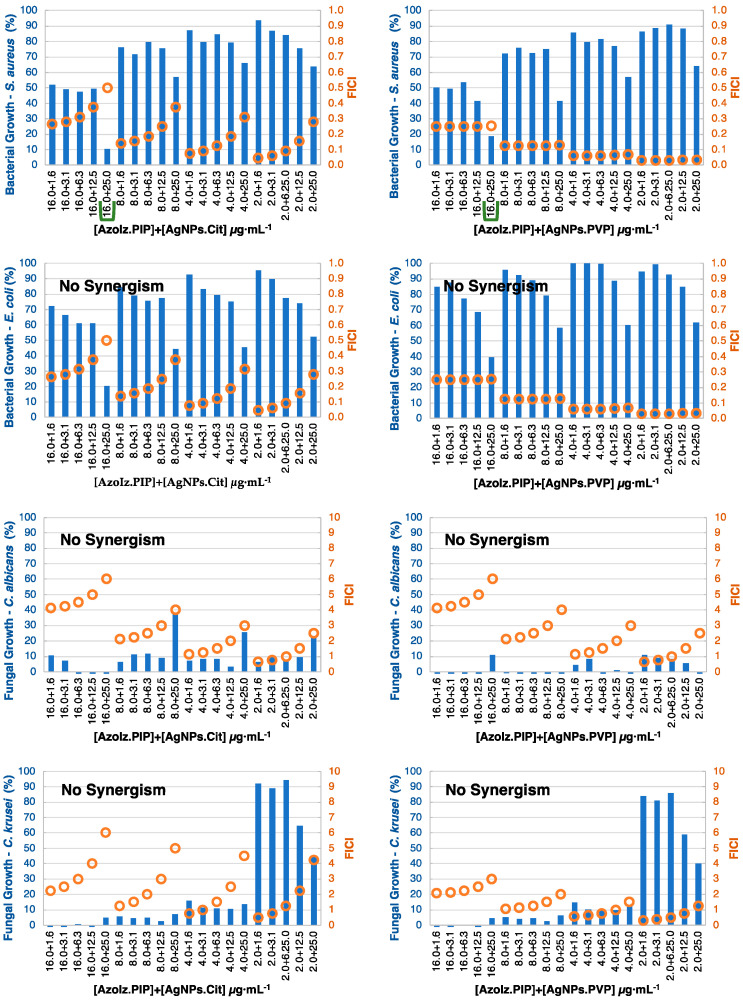
Checkerboard results by combining AzoIz.Pip (concentration ≤ 16.0 μg·mL^−1^) with AgNPs (concentration ≤ 25.0 μg·mL^−1^) showing positive synergistic results (green lines). Bars (blue) represent the microbial growth percentage, and dots (orange) represent the FICI values.

**Figure 9 pharmaceutics-15-00926-f009:**
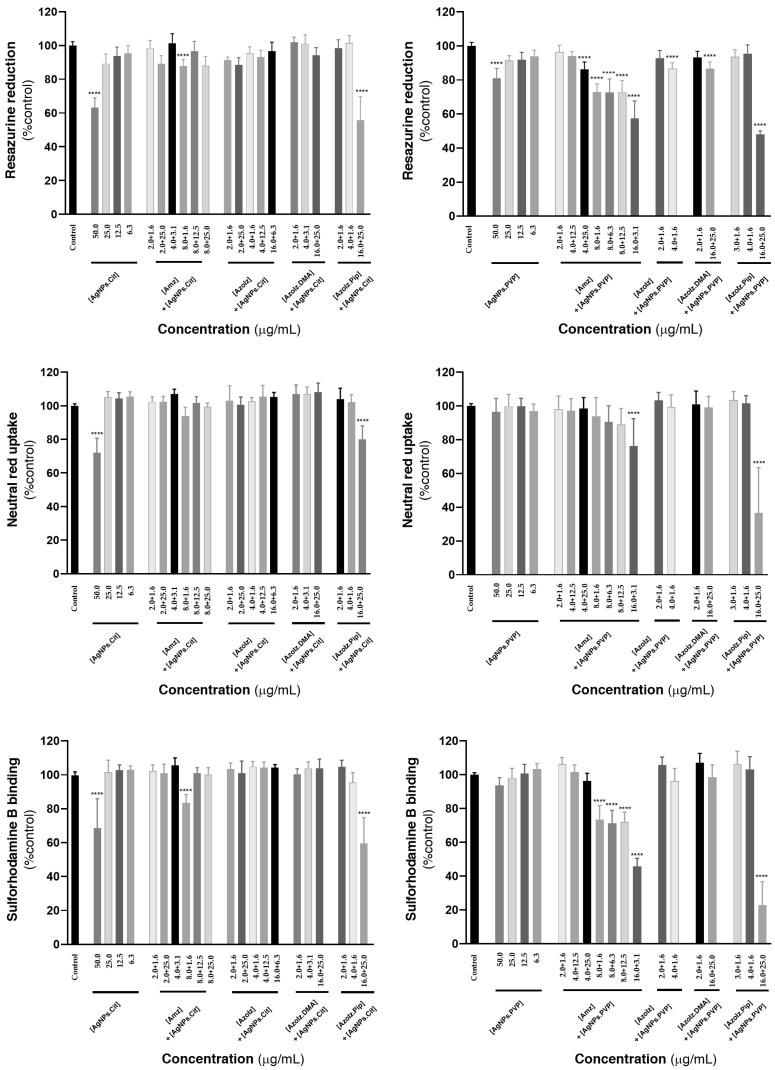
Compounds cytotoxicity towards HaCaT cells, evaluated by the neutral red uptake, resazurin reduction and sulforhodamine B binding assays, 24 h after exposure. Results are expressed as Mean + SD from 4 independent experiments, performed in triplicate. Statistical comparisons were made using one-way ANOVA, followed by Dunnett’s multiple comparisons test (**** *p* < 0.0001 vs. control).

**Table 1 pharmaceutics-15-00926-t001:** Size, PdI, and zeta potential values of AgNPs.Cit or AgNPs.PVP alone, in a concentration of 25.0 μg·mL^−1^, and conjugated with organic molecules in two different concentrations (16 μg·mL^−1^ and 32.0 μg·mL^−1^).

		AgNPs.Cit			AgNPs.PVP		
	[μg·mL^−1^]	Size (nm)	PdI	Zeta Potential (mV)	Size (nm)	PdI	Zeta Potential (mV)
**AgNPs**	25.0	34.89 ± 1.3	0.3	−30.5 ± 0.1	273.2 ± 1.6	0.4	−22.6 ± 0.1
**Amz**	16.0	1104.0 ± 94.6	0.4	0.8 ± 0.2	258.7 ± 23.8	0.3	−8.6 ± 0.3
	32.0	713.7 ± 187.6	0.3	−13.3 ± 0.6	369.0 ± 53.3	0.5	−7.5 ± 0.2
**AzoIz**	16.0	426.6 ± 35.9	0.5	−22.8 ± 0.4	252.6 ± 16.3	0.4	−19.4 ± 0.3
	32.0	748.2 ± 1.1	0.3	−12.7 ± 0.1	345.8 ± 65.8	0.5	−18.0 ± 0.6
**AzoIz.DMA**	16.0	40.85 ± 3.5	0.3	−27.2 ± 0.5	438.0 ± 54.9	0.5	−13.1 ± 0.5
	32.0	39.31 ± 1.2	0.3	−25.3 ± 0.7	527.6 ± 4.7	0.5	−3.4 ± 0.2
**AzoIz.Pip**	16.0	21,740.0 ± 234.6	0.5	−7.5 ± 0.4	16,590.0 ± 174.3	0.7	−1.7 ± 0.6
	32.0	795.6 ± 112.9	0.3	2.3 ± 0.1	496.6 ± 6.7	0.5	0.8 ± 0.4

**Table 2 pharmaceutics-15-00926-t002:** Minimum inhibitory concentrations (MICs) of organic compounds and AgNPs that were used for calculating the fractional inhibitory concentration index (FICI), which was further applied in synergism interpretation.

MIC [μg·mL^−1^]						
	Amz^+^	AzoIz^+^	AzoIz.DMA^+^	AzoIz.Pip^+^	AgNPs.Cit	AgNPs.PVP
** *S. aureus* **	128.0	128.0	64.0	64.0	100.0 [44]	4000.0 [45]
** *E. coli* **	128.0	128.0	128.0	64.0	100.0 [44]	4000.0 [45]
** *C. albicans* **	8.0	4.0	8.0	4.0	12.5	12.5
** *C. krusei* **	16.0	8.0	8.0	8.0	6.3	25.0

**Table 3 pharmaceutics-15-00926-t003:** Summary of the concentrations of organic molecules + AgNPs (μg·mL^−1^) that act in synergy against bacteria and yeasts.

Amz							
*S. aureus*		*E. coli*		*C. albicans*		*C. krusei*	
AgNPs.Cit	AgNPs.PVP	AgNPs.Cit	AgNPs.PVP	AgNPs.Cit	AgNPs.PVP	AgNPs.Cit	AgNPs.PVP
16.0 + 1.6 16.0 + 3.1 16.0 + 6.3 16.0 + 12.5 16.0 + 25.0 8.0 + 12.5 8.0 + 25.0 4.0 + 25.0 2.0 + 25.0	No synergism	16.0 + 1.6 16.0 + 3.1 8.0 + 25 4.0 + 3.1 2.0 + 25.0	16.0 + 3.1 8.0 + 6.3 8.0 + 12.5 4.0 + 12.5 4.0 + 25.0	2.0 + 1.6 2.0 + 3.1	2.0 + 1.6 2.0 + 3.1	No synergism	No synergism
**AzoIz**							
** *S. aureus* **		** *E. coli* **		** *C. albicans* **		** *C. krusei* **	
AgNPs.Cit	AgNPs.PVP	AgNPs.Cit	AgNPs.PVP	AgNPs.Cit	AgNPs.PVP	AgNPs.Cit	AgNPs.PVP
No synergism	No synergism	16.0 + 6.3 4.0 + 12.5 2.0 + 25.0	No synergism	No synergism	No synergism	No synergism	No synergism
**AzoIz.DMA**							
** *S. aureus* **		** *E. coli* **		** *C. albicans* **		** *C. krusei* **	
AgNPs.Cit	AgNPs.PVP	AgNPs.Cit	AgNPs.PVP	AgNPs.Cit	AgNPs.PVP	AgNPs.Cit	AgNPs.PVP
16.0 + 25.0	16.0 + 25.0	No synergism	No synergism	2.0 + 1.6 2.0 + 3.1	2.0 + 1.6 2.0 + 3.1	No synergism	No synergism
**AzoIz.Pip**							
** *S. aureus* **		** *E. coli* **		** *C. albicans* **		** *C. krusei* **	
AgNPs.Cit	AgNPs.PVP	AgNPs.Cit	AgNPs.PVP	AgNPs.Cit	AgNPs.PVP	AgNPs.Cit	AgNPs.PVP
16.0 + 25.0	16.0 + 25.0	No synergism	No synergism	No synergism	No synergism	No synergism	No synergism

## Data Availability

Not applicable.

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
