# Peer review of "Synergistic Antimicrobial Activity of Silver Nanoparticles with an Emergent Class of Azoimidazoles"

_pharmaceutics, 2023, doi:10.3390/pharmaceutics15030926_

Round 1

Reviewer 1 Report

The manuscript is relative good, and appropriate for publication in this journal.  When one reads the abstract and then go to methodology, it was interesting to note that in the methodology, it was indicated that the compounds were characterized by proton NMR, and later it was indicated that atomic absorption spectroscopy was used to determine concentration of Silver.  It was not that clear whether this was referring to the current study or the previous study done by the same group of researchers. If this was done in the current study, the suggestion is that the authors should consider including both NMR and atomic absorption spectroscopy in the abstract as it will indicate the rigorous nature of methods also included in the study.  The authors should also make it clear that these methods were used in the current study if this was the case in the methodology section itself.

Author Response

  1. The manuscript is relative good, and appropriate for publication in this journal.  When one reads the abstract and then go to methodology, it was interesting to note that in the methodology, it was indicated that the compounds were characterized by proton NMR, and later it was indicated that atomic absorption spectroscopy was used to determine concentration of Silver.  It was not that clear whether this was referring to the current study or the previous study done by the same group of researchers. If this was done in the current study, the suggestion is that the authors should consider including both NMR and atomic absorption spectroscopy in the abstract as it will indicate the rigorous nature of methods also included in the study.  The authors should also make it clear that these methods were used in the current study if this was the case in the methodology section itself.

R1: The following information was added to the abstract: “Proton nuclear magnetic resonance was used to confirm the purity of the compounds before further tests and atomic absorption spectroscopy to verify the concentration of silver in the prepared dispersions.” This sentence was added to the materials and methods section: “and the corresponding peaks were compared with the pure products previously characterized (doi: 10.1039/D3CC00372H).” Since the spectra were compared with the previously published results, which are available in the complete characterization of the compounds (including the NMR data), the spectra were not added to the manuscript.

Reviewer 2 Report

Introduction

1. I recommend moving the last paragraph (lines 110 - 129) to section 2. Materials and Methods

 Materials and Methods

2. Please indicate the brands and manufacturers of the instruments used (NMR Spectroscopy, ultrasonic bath…)

Results and discussion

3. The first paragraph of paragraph 3.1 (earlier results, lines 347 - 360) is more appropriate to remove to the end of the Introduction section.

4. Line 373 - probably a typo in the text (Figure 4). The authors talk about absorption bands in ultraviolet and visible regions, however, a link is given to the figure with electron microscopy.

5. It is recommended to improve the quality of figure 3, the signatures are almost unreadable

6. The paper does not explain the choice of average concentrations of 16 μg•mL-1 and 32.0 μg•mL-1 organic molecules for the analysis of the size and zeta potential of AgNPs, despite the fact that for the analysis of the synergistic effect between the organic molecules and AgNPs was tested against microorganisms a wide range of concentrations was taken (128, 64, 32, 16, 8, 4, 2 μg•mL-1).

7. Missing file with additional information.

8. In the tables and charts of paragraph 3.4 there are no indications of statistically significant differences.

9. In general, it is very difficult to understand the results due to the redundancy of data. The authors should formulate the results more clearly.

Author Response

  1. Introduction - I recommend moving the last paragraph (lines 110 - 129) to section 2. Materials and Methods

R1: The paragraph has been moved and the text adapted.

  1. Materials and methods - Please indicate the brands and manufacturers of the instruments used (NMR Spectroscopy, ultrasonic bath…)

R2: The information was added to the manuscript.

  1. Results and discussion - The first paragraph of paragraph 3.1 (earlier results, lines 347 - 360) is more appropriate to remove to the end of the Introduction section.

R3: The paragraph was moved.

  1. Results and discussion - Line 373 - probably a typo in the text (Figure 4). The authors talk about absorption bands in ultraviolet and visible regions, however, a link is given to the figure with electron microscopy.

R4: We apologize for the mistake, all the figure numbers and tables were checked and rectified.

  1. Results and discussion - It is recommended to improve the quality of figure 3, the signatures are almost unreadable.

R5: The font size of the signatures was increased.

  1. Results and discussion - The paper does not explain the choice of average concentrations of 16 μg•mL-1 and 32.0 μg•mL-1 organic molecules for the analysis of the size and zeta potential of AgNPs, despite the fact that for the analysis of the synergistic effect between the organic molecules and AgNPs was tested against microorganisms a wide range of concentrations was taken (128, 64, 32, 16, 8, 4, 2 μg•mL-1).

R6: This technique was used only to verify the trend in the stability of the conjugates with the increase or decrease in the concentration of the molecules since the existing functional groups in the molecules can help or hinder the stability. Thus, lower concentrations have been shown to increase agglomeration, which may be due to low molecule concentration and consequent multiple NPs binding to the same molecule (since the molecules present multiple possible stabilizing moieties). More stabilizing groups are available when the molecule concentration increases and in general better dispersions are obtained. However, studies on the chemical reactivity between molecules and nanoparticles stabilizing agents should be performed, which can be done in future work but are not within the scope of this research. The following sentence was rewritten in the manuscript: “Thus, the maximum concentration of AgNPs was chosen (25.0 mg×mL-1) for all analyses and two middle values of the tested concentrations of organic molecules were combined (16.0 or 32.0 mg×mL‑1).”

  1. Missing file with additional information.

R7: The file was submitted again.

  1. Results and discussion - In the tables and charts of paragraph 3.4 there are no indications of statistically significant differences.

R8: The results did not exhibit a normal distribution. Thus, it was not possible to analyze through parametric statistical analysis. Due to their limited robustness, we decided not to include non-parametric statistical analysis, to prevent misinterpretations.

  1. In general, it is very difficult to understand the results due to the redundancy of data. The authors should formulate the results more clearly.

R9: Some parts were reorganized to avoid redundancy.

Reviewer 3 Report

The manuscript has been reviewed and some corrections need to be required:

1. Polishing in English required.

2. As a part of green synthesis, authors should try to prepare plant based Ag NPs and then carry on further process.

3. Following references should be cited.

https://doi.org/10.1016/B978-0-323-85479-5.00002-2https://doi.org/10.1007/978-981-16-8399-2_6, https://doi.org/10.3390/molecules28020838, https://doi.org/10.1016/B978-0-323-85479-5.00006-X, 

Author Response

  1. Polishing in English required.

R1: The English language was revised.

  1. As a part of green synthesis, authors should try to prepare plant based AgNPs and then carry on further process.

R2: The plant-based synthesis of AgNPs was not in the scope of this research work due to the large number of phytochemicals present in the plant extracts, it would be difficult to perform a systematic control of AgNPs properties. In addition, the phytochemicals could interact with the organic molecules in different ways, making the results difficult to analyze. Since we already have some results using well-defined AgNPs properties, plant-based AgNPs can be considered in future research works.

  1. Following references should be cited.

https://doi.org/10.1016/B978-0-323-85479-5.00002-2, https://doi.org/10.1007/978-981-16-8399-2_6, https://doi.org/10.3390/molecules28020838, https://doi.org/10.1016/B978-0-323-85479-5.00006-X

R3: The references related to the antimicrobial activity of imidazoles were added to the manuscript.